# Evaporation of Saline Droplets on a Superhydrophobic Substrate: Formation of Crystal Shell and “Legs”

**DOI:** 10.3390/ma16145168

**Published:** 2023-07-22

**Authors:** Xiaoqiang Zhang, Hongyue Chen, Zhijun Wang, Nan Wang, Duyang Zang

**Affiliations:** 1MOE Key Laboratory of Material Physics and Chemistry under Extraordinary Conditions, School of Physical Science and Technology, Northwestern Polytechnical University, Xi’an 710129, China; xqzhang@mail.nwpu.edu.cn (X.Z.); chenhongyue@mail.nwpu.edu.cn (H.C.); 2State Key Laboratory of Solidification Processing, Northwestern Polytechnical University, Xi’an 710072, China; zhjwang@nwpu.edu.cn

**Keywords:** superhydrophobic substrate, droplet evaporation, crystallization, internal convection, crystallization pressure

## Abstract

We studied the evaporation-driven crystallization in the droplets of sodium acetate anhydrous (CH_3_COONa) aqueous solution, which were deposited on superhydrophobic substrates. The results reveal distinct crystallization behaviors between saturated and unsaturated droplets under identical experimental conditions. Specifically, unsaturated droplets could form a quasi-spherical crystal shell on the superhydrophobic substrate, while saturated droplets could develop crystal legs between the droplet and substrate when the crystal shell formed. Subsequently, the saturated droplet was lifted off the substrate by the growing crystal legs. The formation of crystal shell was closely associated with the evaporation from the droplet surface and the internal convection inside the droplet. The formation of crystal legs was induced by the heterogeneous nucleation effect caused by the substrate of SiO_2_ nanoparticles. These findings provide valuable insights into regulating the morphology of salt crystallization through adjustments in salt solution concentration and substrate surface structure.

## 1. Introduction

The evaporation of salt solution droplets plays an important role in many applications, such as analytical chemistry [1], coatings [2], biology [3], medical diagnosis [4], and food science [5]. The evaporation process of salt solution droplets involves many physical phenomena, such as contact line movements [6,7], natural and evaporation-induced convection [8,9], and crystallization [10,11]. Subjects related to droplet evaporation have been investigated for years, including the influence of the external environment (temperature, humidity and pressure) on droplet evaporation, the crystallization and phase transition inside the droplets, and the physical or chemical process at the contact lines.

The study of the evaporation of salt droplets under different environmental conditions provides insights into evaporation models and crystallization mechanisms. The flash evaporation process of saltwater droplets in a vacuum chamber showed that the level of salt in solution decreased the intensity of droplet evaporation. With the increase in salt concentration in water, the evaporation rate decreased [12,13]. The evaporation and crystallization processes in the pendant droplet showed that when the evaporation rate was low, a salt shell formed outside the droplet. Once the salt shell formed, the evaporation rate decreased and changed from linear to nonlinear [14]. In contrast, the evaporation of sessile salt droplets is closely related to the contact angle and contact line. G.V. Kuznetsov et al. reported that crystalline hydrates grew near the contact line and the droplet profile was distorted [15]. Virginie Soulié showed that low salt concentrations (≥10^−6^ mol/L NaCl) lead to pinning of the three-phase contact line. Droplets with salt concentration of ≤10^−7^ mol/L NaCl evaporate in a constant contact angle model [16]. These reports indicate that the contact line plays an essential role in the evaporation-caused crystallization of sessile salt droplets.

To minimize the influence of the contact line on droplet evaporation and crystallization, we investigated the process of CH_3_COONa crystallization during droplet evaporation on superhydrophobic substrates. Currently, the superhydrophobic surfaces can be prepared by several different techniques ranging from templating [17], etching [18], and deposition [19] to anodic oxidation [20], etc. Here, we adopted a facile dip-coating method [21]. Our findings indicate that evaporating CH_3_COONa solution droplets on a superhydrophobic substrate leads to the formation of crystal shells and legs. Notably, the temperature required for crystal leg growth in this study (20~30 °C) was significantly lower than that reported in previous research (60~100 °C) for similar phenomena [22,23]. The formation of CH_3_COONa crystal shells and legs, approximately at room temperature, holds significant value in various applications, including the preparation of heteromorphic crystals, corrosion mitigation on solid substrates, and self-transport during the crystallization of salt droplets.

## 2. Materials and Methods

### 2.1. Preparation of Superhydrophobic Substrate

The superhydrophobic substrates were fabricated by immersing a glass slide into a sol containing nano-sized silica (SiO_2_) particles which were synthesized via the Stöber process and doped with hexamethyl disilazane (HMDS) [24].

A certain volume of ethanol (C_2_H_5_OH, ≥99.5%, AR) and tetraethyl orthosilicate (TEOS, C_8_H_20_O_4_Si, 99.99% metals basis) was transferred into a beaker and stirred using a magnetic stirrer (HJ-2B, SHUANGXU, Beijing, China) at a speed of 600~700 r/min for 20~30 min at room temperature. Ammonium hydroxide aqueous solution (NH_4_OH, AR) was subsequently introduced into the mixture and stirred for 20~30 min. The beaker was then sealed and allowed to settle at room temperature for 5~7 days. Finally, hexamethyl disilazane (HMDS, C_6_H_19_NSi_2_, 98%, AR) was added to the prepared mixture and stirred for another 20–30 min. The final mixture was settled at room temperature for 1~2 days, yielding a sol suitable for coating substrates. The volume ratio of reagents used in sol preparation was C_2_H_5_OH:TEOS:NH_4_OH:HMDS = 10:1:0.36:0.56, with all reagents sourced from Aladdin Industrial Corporation, China.

To fabricate the superhydrophobic substrate, a glass slide was impregnated into the sol and stirred for 20~30 s using a dip coater (HTDC-300M, HTLAB, Shanghai, China) with a descending speed of 2 mm/s and an ascending speed of 4 mm/s. After being soaked for 20~30 s and naturally air-dried for 10~15 min at room temperature, the substrate was coated. The surface microstructure of the superhydrophobic substrate was characterized using a field emission scanning electron microscope (FESEM, Zeiss Supra 55, Jena, Germany). Image-Pro Plus 6.0 was used to analyze the SEM images for the quantitative characterization of the substrate.

### 2.2. Preparation of CH_3_COONa Solution

The CH_3_COONa salt solution was prepared by dissolving sodium acetate anhydrous salt (CH_3_COONa, ≥99.0%, AR, Aladdin Industrial Corporation, Shanghai, China) in water and stirring the mixture with a magnetic stirrer at a speed of 600~700 r/min for approximately 30 min. To obtain a saturated sodium acetate salt solution, excess sodium acetate salt was added to water until it could not completely dissolve. After being stirred for 2 h, and with the remaining solid residue filtered out, the saturated solution was obtained. The water used in this study was purified using an ultrapure water system (UPTA-20L, China).

### 2.3. Experimental Procedure

The experimental setup is illustrated in Figure 1. A saline droplet (10 μL) was deposited onto the superhydrophobic substrate using a microsyringe. The evaporation and crystallization process of the droplet were monitored in real time using a high-definition CCD video microscope (GP-640S, GaoPin, Guangzhou, China). The evaporation-induced change in droplet mass was monitored using a high-precision electronic balance (FA1004, Sunny Hengping Instrument, Shanghai, China) within a closed acrylic chamber to minimize the impact of external perturbations such as air flow and contamination. The surface tension of CH_3_COONa aqueous solution with various concentrations was measured using a surface-tension meter (JK99C, Powereach, Shanghai, China). The morphology of crystals and the thickness of crystal shell were characterized using a field emission scanning electron microscope. The experimental temperature was maintained at 25 ± 2 °C while the relative humidity (RH) was kept between approximately 40 and 45%.

## 3. Results and Discussion

### 3.1. Characterization of the Superhydrophobic Substrate

The wettability of the coated substrates was characterized by apparent contact angle (CA) and sliding angle (SA), which were measured using a contact angle meter (SDC-200S, Powereach, China). The results show that a water droplet (10 μL) had a CA of 160 ± 2° (Figure 2a) and an SA of 5° (Figure 2b) on this substrate, indicating that the prepared substrates had an excellent superhydrophobicity. The SEM image showed that SiO_2_ nanoparticles were randomly stacked on the substrate to form a hierarchical micro-nano porous surface (Figure 2c). The SiO_2_ particles on the substrate surface ranged from approximately 100 to 300nm, with an average diameter of 225 nm, as calculated using Gauss fitting (Figure 2c). The density of the SiO_2_ particles was also quantified by Image-Pro Plus 6.0 to be 1.05 × 10^10^/cm^2^. Based on the number (4~5 layers) of accumulated layers of particles and the average diameter (225 nm), the thickness of silica coating measures approximately 1.0 μm. The superhydrophobic coating exhibits a surface roughness of ~0.10 μm, as determined by the formula R_a_ = $\frac{1}{n}\sum \left(\left|Z_{i}\right|\right)$ [25], where R_a_ is the average roughness, and $\left|Z_{i}\right|$ represents the vertical distance from the particles located on the top layer to the mean height.

The superhydrophobicity of the substrate was determined via both the surface chemistry of SiO_2_ nanoparticles and the surface microstructure of the substrate. On the one hand, SiO_2_ nanoparticles were silanized during the preparation of the sol. The introduction of alkyl chains on the surface of SiO_2_ nanoparticles reduces their surface energy significantly. On the other hand, the hierarchical micro-nano porous structures enable the trapping of air pockets which greatly reduce the solid–liquid contact area [26], eventually leading to the Cassie–Baxter state of the deposited droplets [27], as shown in Figure 2d.

### 3.2. Formation of Crystal Shell

As shown in Figure 3a, when a droplet of unsaturated CH_3_COONa solution (10 μL, 20 wt.%) is deposited on the superhydrophobic substrate, it has an approximate spherical shape due to its high contact angle of 160 ± 2° on the superhydrophobic surface. In addition, the size of the deposited droplet is smaller than the capillary length: $l_{c}=\sqrt{\frac{\gamma_{eff}}{\rho g}}$ [28] (*γ_eff_* is surface tension, *ρ* is the density of droplet, *g* is gravitational acceleration). For a 10 μL CH_3_COONa solution droplet, *R*~1.34 mm and $l_{c}$~2.00 mm; thus, the effect of gravity can be ignored in the whole process of droplet evaporation.

**Development of internal natural convection.** Being deposited on the substrate, there is a convection inside the droplet which flows upward along the central axis of the droplet and downward along the droplet surface (Figure 3a). On a superhydrophobic substrate, the evaporation rate is highest at the top of the droplet [29,30]. A higher evaporation rate leads to more solvent loss. As a result, a decreasing concentration gradient from the top to the contact line was observed on the droplet surface. Our experimental results (Figure 4) demonstrate that the surface tension of a CH_3_COONa salt solution decreases with the increasing concentration. Therefore, this decreasing concentration gradient results in an increasing surface tension gradient, which induces a Marangoni flow along the air–liquid interface of the droplet. The Marangoni flow flowed from the top of the droplet to the contact line along the droplet surface. Under the restriction of the substrate, the Marangoni flow increased in the center of the droplet, ultimately generating a symmetrical annular vortex, as shown in Figure 3a and the inset of Figure 4.

**Crystallization on droplet surface.** As evaporation proceeded, there was a significant volumetric loss of the droplet. Some small crystal grains were observed at the top of the droplet surface (Figure 3b). With further evaporation, small crystal grains grew from the top of the droplet downwards along the surface to the contact line (Figure 3c). Eventually, the growing crystal formed a spherical shell that coated the entire droplet surface (Figure 3d). The SEM image of the surface morphology of the spherical crystal shell is shown in Figure 5a–c, which indicates that the crystal shell resulting from the evaporation of CH_3_COONa solution droplet completely differs from the long rod-like structure formed in the CH_3_COONa solution. Figure 5d shows that the thickness of the crystal shell is about 20 μm. Figure 6 shows the mass of the droplet as a function of time during evaporation. After 41.5% of the water in the droplet was evaporated, the concentration of droplet became saturated (34.2 wt.%) at *t* ≈ 1300 s (P point in Figure 6). However, crystal grains at the top of the droplet were actually observed at *t* ≈ 3000 s (Figure 3b and Figure 7b). This can be explained by the Kelvin equation, which elucidates the relationship between concentration of grain-saturated solution and crystal radius via the following equation [31,32,33]:(1)$$RT\ln\frac{c_{2}}{c_{1}}=\frac{2\sigma M}{\rho }(\frac{1}{r_{2}}-\frac{1}{r_{1}})$$
where *c*_1_ and *c*_2_ represent the solubility of grains with radii *r*_1_ and *r*_2_, respectively. *Σ* is the interfacial tension between crystal and solution, *M* is the molar mass, *ρ* is the density of crystal, *R* is the thermodynamic gas constant, and *T* is the absolute temperature.

It should be noted that even the droplet concentration reached critical saturation (34.2 wt.%), no evidenced crystal nuclei formed. As the concentration of the droplet increases to 58.8 wt.%, which is much higher than the critical saturation concentration of 34.2 wt.%, stable grains are observed (Figure 7b,c).

The growth of crystals on the droplet surface is attributed to a combination of factors, including droplet evaporation, droplet surface tension, and internal convection. The evaporation rate at the top of the droplet was the fastest when it was on a superhydrophobic surface. The evaporation rate was lower at a greater distance away from the top of the droplet surface [34,35]. When the surface concentration of the droplet reaches saturation, nucleation occurs first at the top of the droplet. With further evaporation, the nuclei gradually grow to form needle-shaped CH_3_COONa crystals that are buoyant on the droplet surface due to its surface tension. The needle-shaped CH_3_COONa crystals adhere to each other during their growth process, forming a localized crystal shell that acts as a “cap” covering the top surface of the droplet. During this period, the droplet provides support for the crystal “cap” to float on its surface due to surface tension. Furthermore, the internal convection within the droplet would also prevent the crystal shell from sinking until it grows to fully cover the surface of the droplet. The internal convection within the droplet not only generated an upward lifting force, but also provided a constant supply of CH_3_COO^-^ and Na^+^ for the growth of crystals at the droplet surface. Consequently, the CH_3_COONa crystals tend to grow exclusively on the surface of droplets and eventually form a crystal shell, as shown in the corresponding inset of part (Ⅲ) in Figure 5.

### 3.3. Formation of “Legs” for Saturated Droplet

By contrast, the saturated droplet had a distinct phenomenon after the formation of the crystal shell. For the saturated droplet, the substrate acted as heterogeneous nucleation sites immediately once the droplet was deposited on it. The crystals generated via heterogeneous nucleation on the substrate are constantly growing. As a result, when the saturated droplet forms a crystal shell, several crystal legs were observed between the crystal globe and the substrate (Figure 8a(1),a(2)). With further evaporation, the growing crystal legs gradually lifted the crystal globe away from the substrate until the evaporation was completed. It can be observed that the growth of crystal legs was initiated, and their size continuously increased at the points where they contacted the substrate (Figure 8a(3)–a(5)). The peak value of growth rate for a crystal leg reached 0.13mm/min (Figure 8d). The salt solution containing tiny crystals was transported from the crystal globe to the substrate via the growing tubular crystal “leg” (Figure 8b,c). This indicates that the salt crystals carried by the evaporative flow of water inside the hollow crystal legs are critical for their growth.

The emergence of crystal legs is the result of heterogeneous nucleation on the surface of the substrate, which differs from the homogeneous nucleation occurring on the droplet surface. When the droplet contacted the substrate, many annular contact lines formed at the solid–liquid interface due to the rough stacking of silanized SiO_2_ nanoparticles (Figure 9a). These annular contact lines provided heterogeneous nucleation sites for crystallization. For a saturated CH_3_COONa solution droplet, the crystal nuclei inside the droplet preferentially adhere to the annular contact line on the solid–liquid interface due to the heterogeneous nucleation effect (Figure 9b(i)). The continuous accumulation of crystals on the annular contact lines promotes the growth of hollow crystal legs (Figure 9b(ii)).

During the growth of crystal legs, the evaporation at the annular contact lines where the crystal legs contacted the substrate led to capillary compensation flow in the crystal legs [36], as shown in Figure 8c and Figure 9b(ii). This capillary compensation flow continuously transported the salt solution from crystal globe to the substrate through the hollow crystal legs, which provided a constant supply of raw materials (CH_3_COONa grains) for the growth of crystal legs.

The superhydrophobic substrate has a rough surface with lots of micro- and nano-sized porous structures formed by silanized SiO_2_ nanoparticles. Crystal growth in porous media can induce crystallization pressure [22]. Therefore, crystallization pressure is a crucial factor during the growth of crystal legs. Crystallization pressure is described by the following equation [22,37,38]:(2)$$P_{crys}=\frac{vRT}{V_{m}}\left(\ln(S)+\ln\frac{\gamma_{\pm }}{\gamma_{\pm ,0}}+\frac{v_{0}}{v}\ln\frac{a_{w}}{a_{w}{}_{,0}}\right)$$
where *v* is the number of different ions released upon the complete dissociation of the salt (for CH_3_COONa, *v* = 2), *R* is the thermodynamic gas constant, *T* is the absolute temperature, *V_m_* is the molar volume of the solution, *S* is the supersaturation, *γ*_±_ is the mean activity coefficient, and *a_w_* is the water activity. The activity coefficient reflects the difference between the effective concentration and the actual concentration. The higher the concentration of the solution, the lower the activity. For saturated CH_3_COONa solution, its activity can be ignored. In the practical calculation, Equation (2) can be simplified as [39,40]:(3)$$P_{crys}\approx \frac{vRT}{V_{m}}\ln(S)$$

In Equation (3), the crystallization pressure *P_crys_* was calculated to be 447.9~457.2 Pa, which is much higher than the hydrostatic pressure of the droplet (~29.02 Pa) This suggested that the crystallization pressure was large enough to overcome the gravity of crystal globe and lifted it off the substrate. Therefore, the crystallization pressure generated during crystallization of sodium acetate crystals is large enough to act as the driving force for the growth of crystal legs.

### 3.4. Discussion

The emergence of crystal legs is mainly due to the heterogeneous nucleation effect of the superhydrophobic substrate on the crystallization of saturated CH_3_COONa droplets. However, the final crystallization results of saturated and unsaturated CH_3_COONa solution droplets on the same substrate dramatically differed. This is because, at the beginning of saturated CH_3_COONa solution droplet evaporation, both the surface and interior of the droplets are already saturated. Due to the lower nucleation barrier of heterogeneous nucleation compared to that of homogeneous nucleation, crystal nuclei tend to preferentially attach and grow on rough surfaces. Consequently, during the evaporation of a saturated droplet to form a crystal shell, the heterogeneous nucleation effect results in the formation of crystal legs between the droplet and the substrate, which subsequently lift it off from the substrate.

For the droplet of unsaturated solution, the crystallization was indeed dominated by homogeneous nucleation, which occurred at the top surface of the droplet [41]. The heterogeneous effect of substrate was inhibited because the evaporation-driven natural convection also promoted the enrichment of solute at the top surface. Finally, when the evaporation was finished, only a crystal shell formed with a hollow inside.

To demonstrate that the emergence of CH_3_COONa crystal legs was induced by a heterogeneous nucleation effect on crystals via the superhydrophobic substrate consisting of SiO_2_ nanoparticles, similar evaporation experiments were conducted on lotus leaves. The results show that both unsaturated and saturated CH_3_COONa solution droplets formed crystal shells after evaporation, but neither grew crystal legs on the surface of lotus leaves (Figure 10). This indicates that there was no heterogeneous nucleation on the surface of lotus leaf during the evaporation and crystallization of CH_3_COONa solution droplets.

The heterogeneous nucleation effect results in diverse crystal morphologies on two distinct surfaces, primarily attributed to the coherent relationship between the nascent phase and the interface of the substrate [42,43,44]. The crystal structure of anhydrous sodium acetate adopts a hexagonal closest-packing arrangement, featuring four tetrahedral voids within each unit cell, while silica exhibits a centroid regular tetrahedron in its spatial arrangement. These two crystal structures are spatially complementary to some extent. Considering this, the nucleation and crystallization of CH_3_COONa are more likely to occur on the surface of SiO_2_ nanoparticles. Additionally, the accumulation of SiO_2_ nanoparticles on the superhydrophobic surface results in the formation of granular layers that resemble the arrangement of unit cell during CH_3_COONa crystal formation. Thus, the CH_3_COONa crystals are inclined to nucleate and crystallize on the protruding annular contact lines, which were formed by SiO_2_ nanoparticles, and eventually form tubular crystal legs, whereas the surface structure of the lotus leaf consists of three parts, including papillae, papillae cover, and wax layer [45,46]. The epicuticular wax is primarily composed of a mixture of aliphatic compounds, particularly nonacosanol and nonacosanediols [47]. These aliphatic compounds exhibit an amorphous structure that is distinct from the crystal structure of CH_3_COONa crystal, which significantly deviates from the regular cubic crystal structure of CH_3_COONa salt. Consequently, the CH_3_COONa crystal nuclei would not attach to the surface of lotus leaf for formation and growth. This indicates that the surface of the lotus leaf has no heterogeneous nucleation effect on the crystallization of CH_3_COONa salt in small droplets.

## 4. Conclusions

In this study, we investigated the process of evaporative crystallization of CH_3_COONa solution droplets on a superhydrophobic substrate formed by silanized SiO_2_ nanoparticles. With the completion of evaporation, unsaturated CH_3_COONa solution droplets formed a spherical crystal shell. Meanwhile, a saturated CH_3_COONa solution droplet was lifted off the substrate by the growing crystal legs underneath the already formed crystal shell. The droplet evaporation, droplet surface tension, and internal convection of droplets are the main reasons for the formation of CH_3_COONa crystal shells. The heterogeneous nucleation effect of the substrate is responsible for the formation of crystal legs for the saturated CH_3_COONa solution droplets. The crystallization pressure of saturated CH_3_COONa solution droplets (447.9~457.2 Pa) was large enough to overcome its hydrostatic pressure (~29.02 Pa) and lift the crystal globe off the superhydrophobic substrate. Evaporation at the annular contact lines led to capillary compensation flow inside the tubular crystal legs, which brought liquid from crystal globe to the contact lines. These results may be reproducible for the morphological control of crystallization associated with droplet evaporation.

## Figures and Tables

**Figure 1 materials-16-05168-f001:**
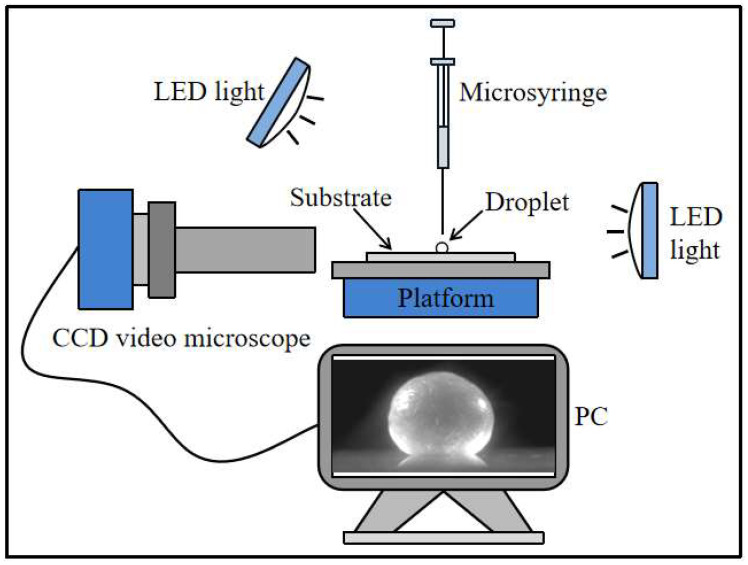
Illustration of the experimental setup for droplet evaporation.

**Figure 2 materials-16-05168-f002:**
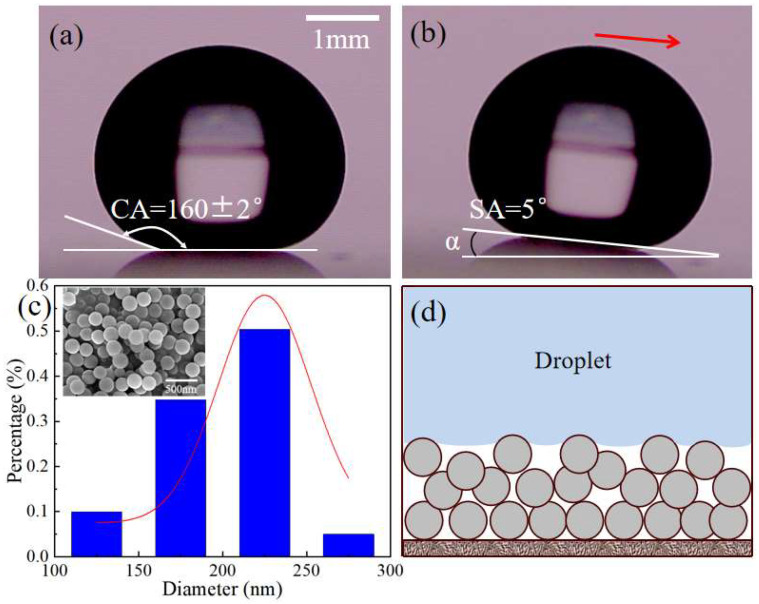
Characterization of the superhydrophobic substrate. (**a**) The apparent contact angle of a water droplet, 160 ± 2°; (**b**) The sliding angle, 5°; (**c**) SEM image of the superhydrophobic surface and SiO_2_ particle size distribution; (**d**) Illustration showing the contact between the droplet and the substrate.

**Figure 3 materials-16-05168-f003:**
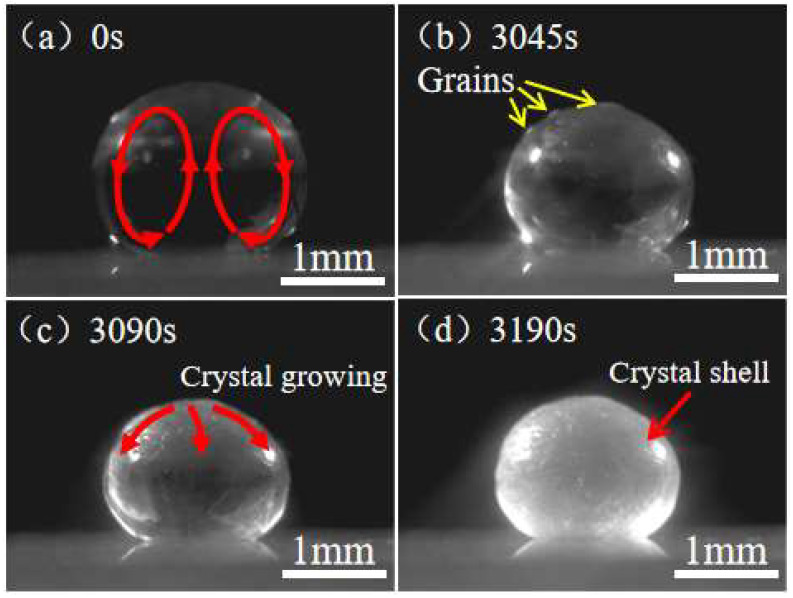
Evaporation-induced crystallization of CH_3_COONa in an unsaturated droplet (20 wt.%) on nanoparticle superhydrophobic substrate.

**Figure 4 materials-16-05168-f004:**
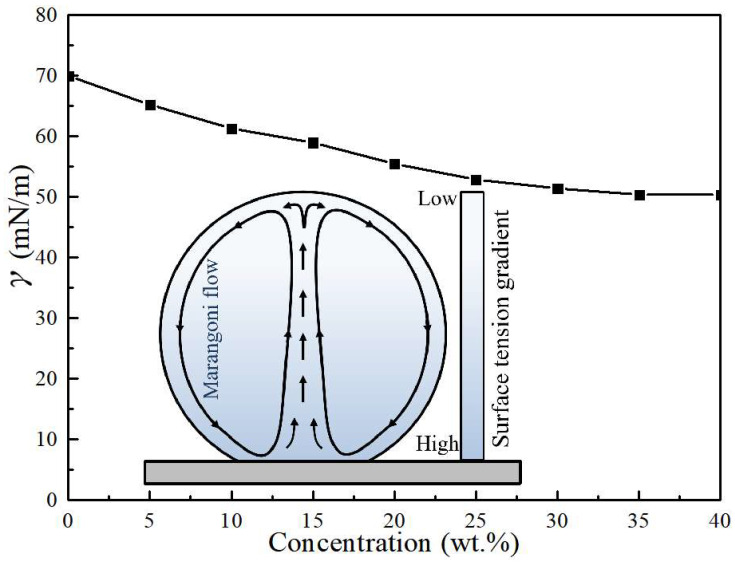
Surface tension of sodium acetate solution as a function of concentration and the illustration of internal flow of a CH_3_COONa solution droplet.

**Figure 5 materials-16-05168-f005:**
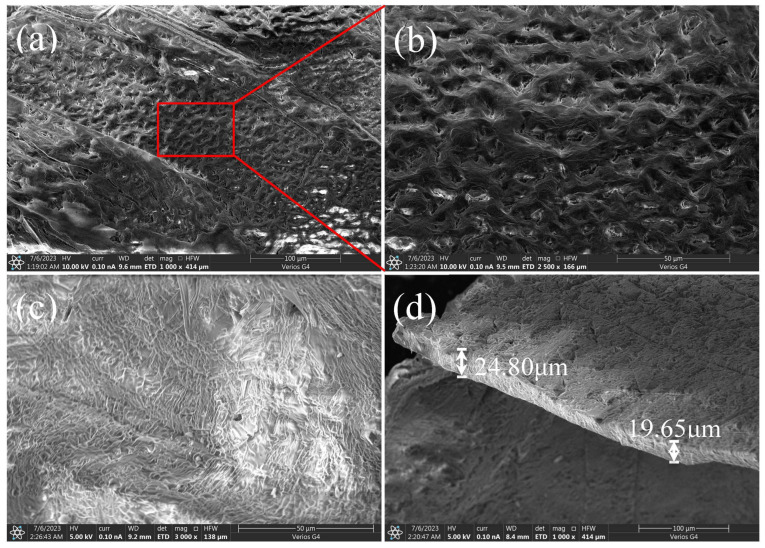
Surface morphology of spherical crystal shell. (**a**) External surface and (**b**) its local magnification; (**c**) internal surface; (**d**) the cross section of the crystal shell.

**Figure 6 materials-16-05168-f006:**
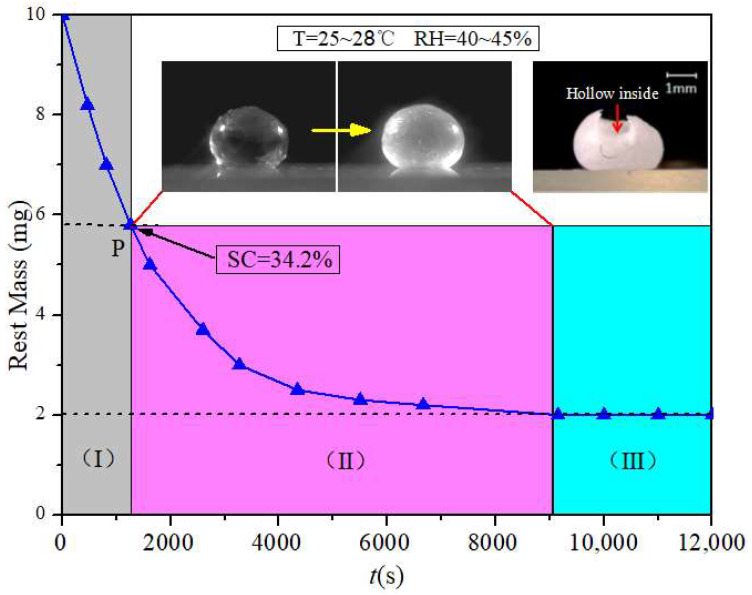
Rest mass–time curve of the droplet during the evaporation and crystallization of a droplet from an unsaturated CH_3_COONa solution to form a crystal shell.

**Figure 7 materials-16-05168-f007:**
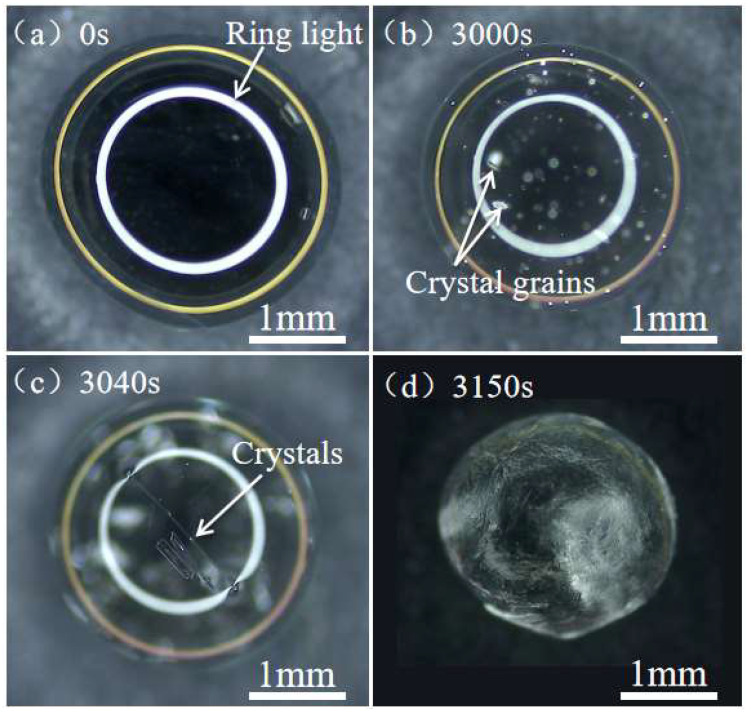
Top view of a CH_3_COONa solution droplet (20 wt.%) evaporating and crystallizing on a nanoparticle superhydrophobic substrate.

**Figure 8 materials-16-05168-f008:**
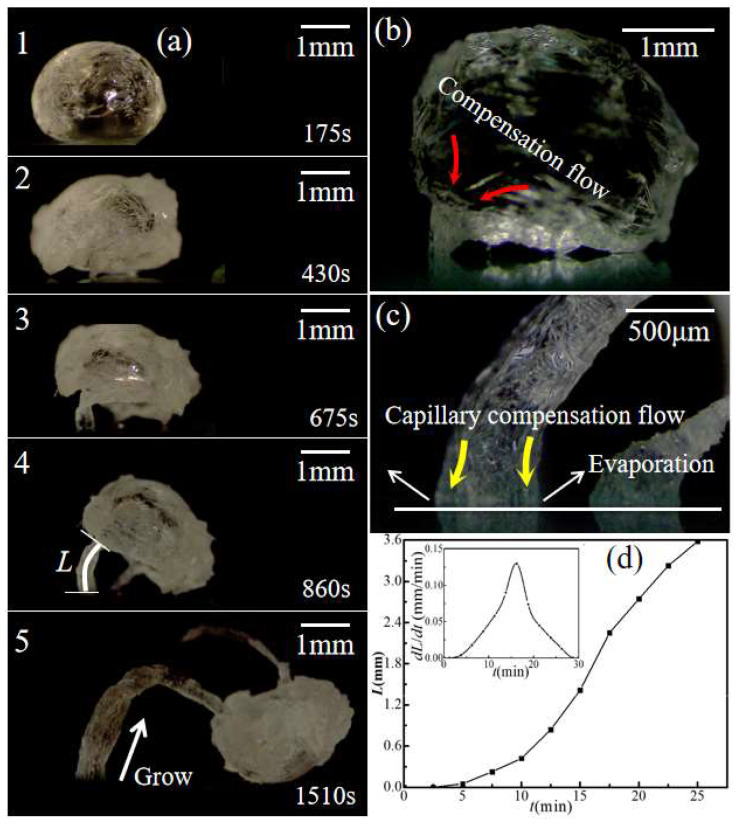
Evaporation-induced crystallization of CH_3_COONa from a saturated droplet (34.2 wt.%) on nanoparticle superhydrophobic substrate. (**a**) Formation of crystal globe and growing legs; (**b**) compensating flow from the droplet to the crystal leg; (**c**) liquid flow inside the crystal leg and evaporation at the annular contact line; (**d**) growth rate of a crystal leg.

**Figure 9 materials-16-05168-f009:**
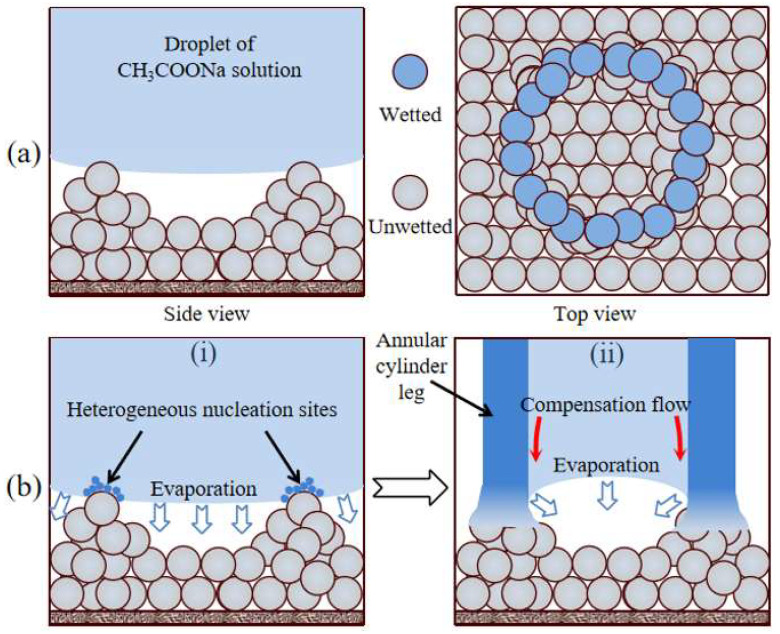
The mechanism responsible for the formation of CH_3_COONa crystal “legs”. (**a**) Illustration showing the annular contact line formed due to the rough stacking of nanoparticles; (**b**) The annular contact line, which led to the heterogeneous nucleation sites provided by the formation of annular cylinder crystal “legs”.

**Figure 10 materials-16-05168-f010:**
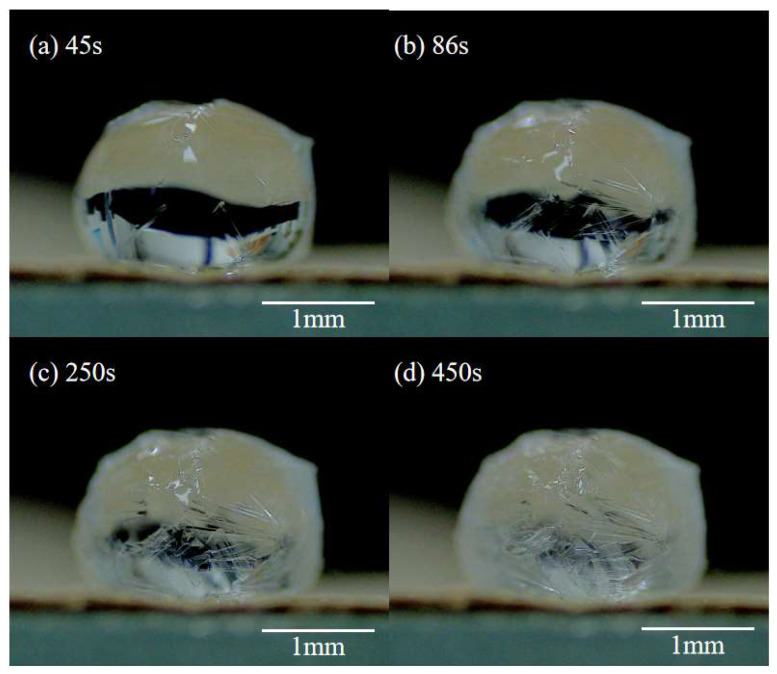
Evaporation and crystallization process of saturated CH_3_COONa solution droplet on lotus leaf.

## Data Availability

All the research data have been presented in the paper.

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
