# Peer review of "Evaporation of Saline Droplets on a Superhydrophobic Substrate: Formation of Crystal Shell and “Legs”"

_materials, 2023, doi:10.3390/ma16145168_

Round 1

Reviewer 1 Report

The paper under review deals with the research on the evaporation-driven crystallization in the droplets of sodium acetate anhydrous (CH3COONa) aqueous solution which were deposited on superhydrophobic substrates. The article tackles an important issue in mass and heat transfer. The authors showed the results of the studies which showed that of crystallization behaviors between saturated and unsaturated droplets under identical experimental conditions can be distinct. These findings provide valuable insights into regulating the morphology of salt crystallization through adjustments in salt solution concentration and substrate surface structure.The authors showed the results of the studies which, I hope, provide guidance to improve analyzed process. The article contains crucial information regarding the research models, methods and experimental techniques.

I am aware that the authors had to select material from an extremely extensive research topic. However, this means that this study is very original. The layout of the article is correct, both in terms of content and didactics. The article has been edited very carefully, and the style and linguistic correctness of the study are extremely correct. While reading it, I did not find any formal or content-related flaws, and any minor editorial errors that I noticed were reported below. The figures included in the study fully reflect the essence of the analysed problems and allow for a complete understanding of the presented content.

Short comments:

1. The state of the art should be presented clearly with a description of a particular publication’s contribution to its development. The collective citations in: page 2 [1-5] and [6-11], are not acceptable in a scientific journal.

2. Minor editorial errors: missing spaces, wrong paragraph, etc. I suggest re-reading the work and correcting any editorial errors.

3. Figure 3, 6 and 8: Drawing scale placed on only one drawing. In my opinion, it should be applied to each drawing separately.

The article contains adequate and appropriately selected 40 literature items and 8 figures. In opinion of the reviewer the article needs minor corretion/complete and than it can be published in the Materials.

Reviewer 2 Report

The manuscript by Zhang et al. reports experimental results on the evaporation-driven crystallization of sodium acetate (on superhydrophobic substrates) from the saturated and unsaturated solutions. Different mechanisms of nucleation for saturated and unsaturated solutions are reported and discussed. The manuscript is well written and the conclusions are supported by the results. I recommend publication of this manuscript subject to the following minor revisions:

1-Page 9, it is written ‘With further evaporation, only when the droplet concentration was much higher than the critical saturation concentration can stable crystal nuclei be generated and grew continuously’. Homogeneous nucleation occurs when the concentration is slightly higher than the critical saturation concentration. If the concentration is much higher than the critical saturation concentration, the nucleation and growth occur quickly (out of equilibrium). In this case there might be a possibility of observing several crystalline domains in the system. Is there any evidence for this? Of course, of different crystalline nuclei, the smaller ones will melt back to the liquid and recrystallize at the surface of bigger one, provided that the nucleation in not highly far from equilibrium (please see J. Chem. Theory Comput. 2022, 18, 1870).

2-For the saturated solution, evaporation supersaturates the top layer. Then why the nucleation/crystallization does not begin from the top layer (similar to that of unsaturated solution)? I understand that formation of crystal legs, attached to the surface, is discussed in the text. However, if heterogeneous nucleation occurs at the surface, how does evaporation assist nucleation at the surface?

The manuscript by Zhang et al. reports experimental results on the evaporation-driven crystallization of sodium acetate (on superhydrophobic substrates) from the saturated and unsaturated solutions. Different mechanisms of nucleation for saturated and unsaturated solutions are reported and discussed. The manuscript is well written and the conclusions are supported by the results. I recommend publication of this manuscript subject to the following minor revisions:

1-Page 9, it is written ‘With further evaporation, only when the droplet concentration was much higher than the critical saturation concentration can stable crystal nuclei be generated and grew continuously’. Homogeneous nucleation occurs when the concentration is slightly higher than the critical saturation concentration. If the concentration is much higher than the critical saturation concentration, the nucleation and growth occur quickly (out of equilibrium). In this case there might be a possibility of observing several crystalline domains in the system. Is there any evidence for this? Of course, of different crystalline nuclei, the smaller ones will melt back to the liquid and recrystallize at the surface of bigger one, provided that the nucleation in not highly far from equilibrium (please see J. Chem. Theory Comput. 2022, 18, 1870).

2-For the saturated solution, evaporation supersaturates the top layer. Then why the nucleation/crystallization does not begin from the top layer (similar to that of unsaturated solution)? I understand that formation of crystal legs, attached to the surface, is discussed in the text. However, if heterogeneous nucleation occurs at the surface, how does evaporation assist nucleation at the surface?

Reviewer 3 Report

This paper describes the evaporation of saline (CH3COONa) droplets on superhydrophobic substrates. Basically, this paper describes the qualitative analysis of the formation of crystal shell and “legs”. Quantitative information and characterization of the crystals that were formed is mandatory in a scientific paper. Therefore, quantitative characterization of the evaporation process is critical (to be abe to reproduce in other laboratories). Also, the crystals should be observed and characterized. Other natural and synthetic surfaces should be assayed.

Round 2

Reviewer 3 Report

This is a revised version of the raw paper. The authors did not provide significant revision because they did not give further scientific information about the solid materials formed neither about the prepared superhydrophobic surface. My previous comment "quantitative characterization of the evaporation process is critical (to be abe to reproduce in other laboratories). Also, the crystals should be observed and characterized." was not answered. Also, further characterization of the superhydrophobic surface should have been done, namely the thickness of the silica coating, the size of the silica particles, the density of the silica particles, the roughness of the coating, etc.
